# Perspective on CO$_2$ Hydrogenation for Dimethyl Ether Economy

Chang Liu and Zhongwen Liu *

Key Laboratory of Syngas Conversion of Shaanxi Province, School of Chemistry and Chemical Engineering, Shaanxi Normal University, Xi'an 710119, China
* Correspondence: zwliu@snnu.edu.cn

**Abstract:** The CO$_2$ hydrogenation to dimethyl ether (DME) is a potentially promising process for efficiently utilizing CO$_2$ as a renewable and cheap carbon resource. Currently, the one-step heterogeneous catalytic conversion of CO$_2$ to value-added chemicals exhibits higher efficiency than photocatalytic or electrocatalytic routes. However, typical catalysts for the one-step CO$_2$ hydrogenation to DME still suffer from the deficient space–time yield and stability in industrial demonstrations/applications. In this perspective, the recent development of the one-step CO$_2$ hydrogenation to DME is focused on different catalytic systems by examining the reported experimental results and the reaction mechanism including the catalytic nature of active sites, activation modes and of CO$_2$ molecules under relevant conditions; surface intermediates are comparatively analyzed and discussed. In addition to the more traditional Cu-based, Pd-based, and oxide-derived bifunctional catalysts, a further emphasis is given to the characteristics of the recently emerged In$_2$O$_3$-based bifunctional catalysts for the one-step conversion of CO$_2$ to DME. Moreover, GaN itself, as a bifunctional catalyst, shows over 90% DME selectivity and a reasonably high activity for one-step CO$_2$ hydrogenation, and the direct hydrogenation of CO$_2$ via the unique non-methanol intermediate mechanism is highlighted as an important illustration for exploring new catalytic systems. With these analyses and current understandings, the research directions in the aspects of catalysis and DME economy are suggested for the further development of one-step DME synthesis from CO$_2$ hydrogenation.

**Keywords:** CO$_2$ hydrogenation; dimethyl ether; bifunctional catalyst; copper; gallium nitride

## 1. Introduction

The increasingly severe issue caused by the cumulative over-emission of greenhouse gas (GHG) is one of the most prominent concerns in the world. Among commonly concerned GHGs, carbon dioxide (CO$_2$), which is mostly derived from the unlimited utilization of fossil fuels, is believed to be a main contributor to the greenhouse effect, global warming and climate change [1–3]. Therefore, reducing the total emission of CO$_2$ becomes an important solution to deal with possible climate change in future.

The capture, storage and utilization (CCSU) CO$_2$ has been considered an effective method to alleviate the problem [4–7]. Chemical conversion of CO$_2$ to high-value-added products exhibits dramatic efficiency in economic and industrial sectors, and has been paid increasing attention. Because the CO$_2$ molecule exhibits high thermodynamic stability but can be employed as a weak oxidant [8,9], conversion of CO$_2$ through hydrogenation becomes an ideal procedure [1,2,10]. Moreover, considering current techniques, the heterogeneous hydrogenation of CO$_2$ should be a more promising technique due to its higher catalytic activity than photocatalytic or electrocatalytic techniques.

In the procedure of heterogeneous CO$_2$ hydrogenation, various products can be obtained. These products can be divided into three types: (i) carbon monoxide (CO), formed from a reversed water–gas shift (RWGS) reaction; (ii) hydrocarbons (HCs), including methane, lower olefins, gasolines, diesels, aromatics, etc.; (iii) oxygenates, including

methanol, dimethyl ether (DME), higher alcohols ($C_2^+$ alcohols), etc. Among them, CO is an intermediate product that needs to be further converted to other products via Fischer–Tropsch synthesis. Moreover, when atom efficiency is taken into account, one of the oxygen atoms in $CO_2$ can remain during the $CO_2$ hydrogenation to oxygenates, indicating that the atom efficiency for producing oxygenates is better than producing HCs.

In the case of the $CO_2$ hydrogenation to oxygenates, single-carbon oxygenates, methanol and DME have been paid more attention due to their higher selectivity than $C_2^+$ alcohols. Previously, Olah et al. [11–13] proposed an industrial carbon cycle based on producing and converting methanol. They believed that in this process, the efficient utilization of $CO_2$ could be realized by the thermochemical or electrochemical hydrogenation with the hydrogen generated from the electrolysis of water. As the intermediate product (Figure 1), methanol can be further converted into high-value-added chemicals, e.g., lower olefins and aromatics. Moreover, in spite of its toxicity, liquid methanol is feasible in the transportation sector; thus, it can also be employed as fuel for fuel cells and as substitute to gasoline due to its high octane rating.

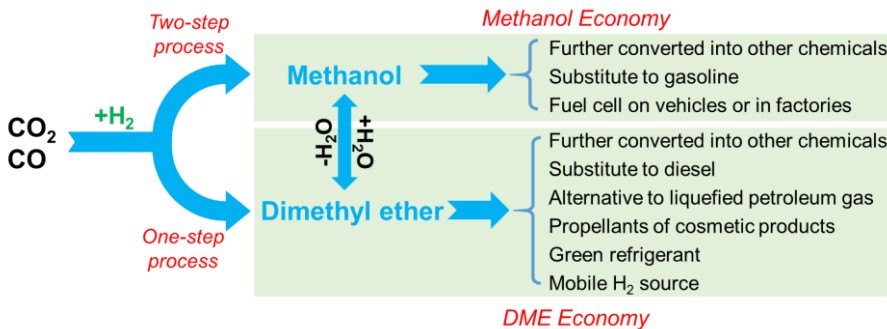

**Figure 1.** Scheme for the formation and utilization of methanol and dimethyl ether.

However, DME can also be employed as an intermediate product of $CO_2$ hydrogenation (Figure 1). During the process of $CO_2$ hydrogenation, formation of DME exhibits a higher energetic efficiency than formation of HCs or alcohols [14], e.g., 97% of energy is stored in DME during $CO_2$ hydrogenation. Moreover, DME is a non-toxic, non-carcinogenic and non-corrosive chemical applied as a propellant of cosmetic products. DME is also employed as a green refrigerant due to its low global warming potential, and as a promising ultra-clean fuel alternative to liquefied petroleum gas and diesel due to its high cetane rating [15]. More importantly, for both CO hydrogenation and $CO_2$ hydrogenation, producing DME is thermodynamically more favorable than methanol. These advantages demonstrate that as the intermediate in the $CO_2$ hydrogenation, DME exhibits more potential than methanol. Therefore, a new chemical economy centered on synthesizing and converting DME, the so-called DME economy [16], is expected to be significantly developed in the chemical industry in the future.

Thus, it is necessary and important to improve the technique of $CO_2$ hydrogenation to DME. There are two kinds of processes to realize the $CO_2$ hydrogenation to DME (Figure 1): (i) two-step process, in which $CO_2$ is firstly hydrogenated to methanol in one reactor, and the methanol is further converted to DME in another reactor; (ii) one-step process, in which $CO_2$ is hydrogenated to DME in the one and only reactor. However, irrespective of the one-step or two-step process [17,18], methanol is employed as the intermediate and two kinds of active sites are employed to catalyze the $CO_2$ hydrogenation to methanol and, subsequently, methanol to DME. Thus, in the one-step $CO_2$ hydrogenation to DME, these two active sites are coupled in one bifunctional catalyst, while the two-step $CO_2$ hydrogenation to DME can be regarded as a special example of one-step $CO_2$ hydrogenation to DME without any interaction between the two active sites. However, considering the total cost and complexity of the process, the one-step $CO_2$ hydrogenation to DME is a more promising process in the DME economy.

According to current research [19–22], the design of an effective bifunctional catalyst is the most important challenge in one-step $CO_2$ hydrogenation to DME, and a great deal of effort has been made. Over the bifunctional catalysts, metal-based catalysts are usually employed for $CO_2$ hydrogenation to methanol, and the reduced Cu is the most studied active site. For the acidic sites catalyzing the dehydration of methanol to DME, different kinds of zeolites are the most employed under the atmosphere of $CO_2$ hydrogenation. Based on these results, several reviews [17,19,23–26] have been published to provide conclusions and perspectives about $CO_2$ hydrogenation to DME. Most of publications focus on the effects of promoters or supports on the catalytic performance. Moreover, the preparation method [20,23,24,27], the design of reactors [17,19,21,24,26] and thermodynamic considerations [18,19,23] are paid attention to, while deactivation and regeneration of catalysts are also involved [28]. To further reveal the effect on the catalytic performance, the nature of active sites, reaction mechanism and acidity of the acidic sites have been investigated [18,21,29,30]. However, in the previous publications, various competing conclusions are obtained and a general agreement needs to be reached.

Due to the advantage of the DME economy and the challenge in the design of an effective catalyst for $CO_2$ hydrogenation to DME, this paper summarizes the recent developments in the investigation of one-step hydrogenation to DME. A thermodynamic approach is presented first. Then, we focus on the nature of active sites and the reaction mechanism over different types of catalysts, e.g., Cu-based catalysts, precious metal-based catalysts, oxide-based catalysts and GaN-based catalysts. According to the previous results, conclusions about the active nature and reaction mechanism over different catalysts are proposed. Based on these analyses, future perspectives are proposed.

## 2. Thermodynamic Study

In the process of $CO_2$ hydrogenation to DME, methanol is usually present as an intermediate product. Moreover, CO is also formed as a principal by-product, and it can also be hydrogenated to methanol or DME. Therefore, thermodynamic properties of different reactions are studied, and results are listed in Table 1. It is observable that the RWGS reaction is endothermic, while the hydrogenation and dehydration reactions are exothermic. Consequently, the increasing reaction temperature significantly inhibits the DME selectivity and improves the CO selectivity, resulting in the limitation of the formation of DME. To achieve highly selective conversion of $CO_2$ to DME, the operation temperature is usually kept low, e.g., <300 °C for the metallic catalysts [30] and <400 °C for the oxide- or GaN-based catalysts [31–33]. However, low reaction temperature also lowers catalytic activity in $CO_2$ hydrogenation to DME. Therefore, improving the catalytic activity by reasonable design of the catalyst is an important challenge in $CO_2$ hydrogenation to DME.

**Table 1.** Thermodynamic results of reactions involved in the $CO_2$ hydrogenation to DME. Results are originated from [18,21,34].

| Reaction | $\Delta H_{298K}$ (kJ·mol$^{-1}$) | $\Delta G_{298K}$ (kJ·mol$^{-1}$) |
|---|---|---|
| (CO$_2$-to-methanol)<br>$CO_2 + 3H_2 = CH_3OH + H_2O$ | −49.4 | +3.5 |
| (CO$_2$-to-DME)<br>$2CO_2 + 6H_2 = CH_3OCH_3 + 3H_2O$ | −122.0 | −5.0 |
| (RWGS)<br>$CO_2 + H_2 = CO + H_2O$ | +41.2 | −28.6 |
| (CO-to-methanol)<br>$CO + 2H_2 = CH_3OH$ | −90.6 | −25.2 |
| (CO-to-DME)<br>$2CO + 4H_2 = CH_3OCH_3 + H_2O$ | −102.4 | −33.5 |

The thermodynamic results also demonstrate that the $CO_2$-to-DME reaction exhibits lower enthalpy and lower Gibbs free energy than the $CO_2$-to-methanol reaction. Shen et al. [35] calculated the equilibrium yield of DME, CO and methanol in the $CO_2$ hydrogenation reaction (Figure 2) and found that under the same reaction conditions, the equilibrium yield of DME is much higher than that of methanol in the $CO_2$ hydrogenation process. This result is consistent with other publications [34,36–40], and authors also indicate that the formation of DME can be improved by high pressure and low temperature. Based on the thermodynamic parameters, Catizzone et al. [41] studied the $CO_2$ equilibrium conversion for $CO_2$ hydrogenation, and found that $CO_2$ hydrogenation to DME exhibited higher catalytic activity than $CO_2$ hydrogenation to methanol in the temperature range of 160–320 °C and pressure range of 10–100 MPa. However, when the reaction temperature further increases (>350 °C), there is only small difference between the selectivity of DME and methanol. Moreover, when partial pressure of $H_2$ becomes lower, $CO_2$ hydrogenation to DME exhibits a higher catalytic activity than $CO_2$ hydrogenation to methanol [34], indicating that the consumption of $H_2$ can be lower for $CO_2$ hydrogenation to DME. Therefore, these results demonstrate that $CO_2$ hydrogenation to DME shows advantages in the thermodynamic parameters, compared with $CO_2$ hydrogenation to methanol.

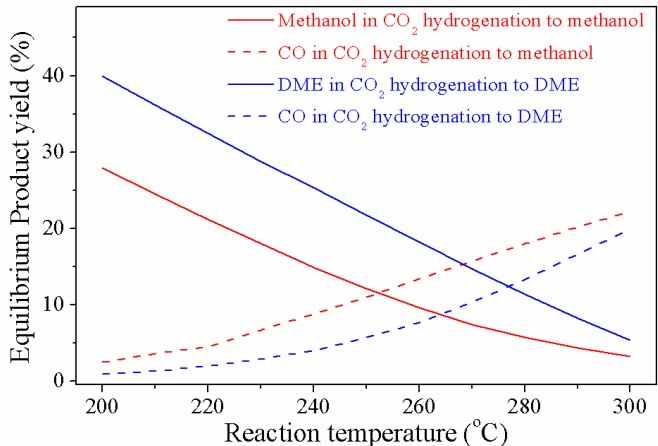

**Figure 2.** Equilibrium product yield in the $CO_2$ hydrogenation. Reproduced with permission from Shen et al. [35], published by Springer Nature, 2000.

One-step $CO_2$ hydrogenation to DME should be more efficient than two-step $CO_2$ hydrogenation to DME. In the case of the one-step procedure, the thermodynamic results are the same as those of the $CO_2$-to-DME reaction shown in Figure 2. In the case of the two-step procedure, we can assume that the equilibrium conversion is achieved in each reactor, then the final equilibrium thermodynamic results can be calculated. The thermodynamic results of methanol dehydration to DME are shown in Figure 3a. We observe that in the methanol dehydration [42], the methanol conversion slightly decreases with increasing reaction temperature, while the DME selectivity remains at nearly 100%. Further, the equilibrium DME yields for the one-step and two-step $CO_2$ hydrogenation to DME are calculated, shown in Figure 3b. It is observable that the DME equilibrium yield in the one-step procedure is higher than that in the two-step procedure, indicating that the one-step $CO_2$ hydrogenation to DME is thermodynamically favorable. Therefore, it could be a promising technique in the future DME economy.

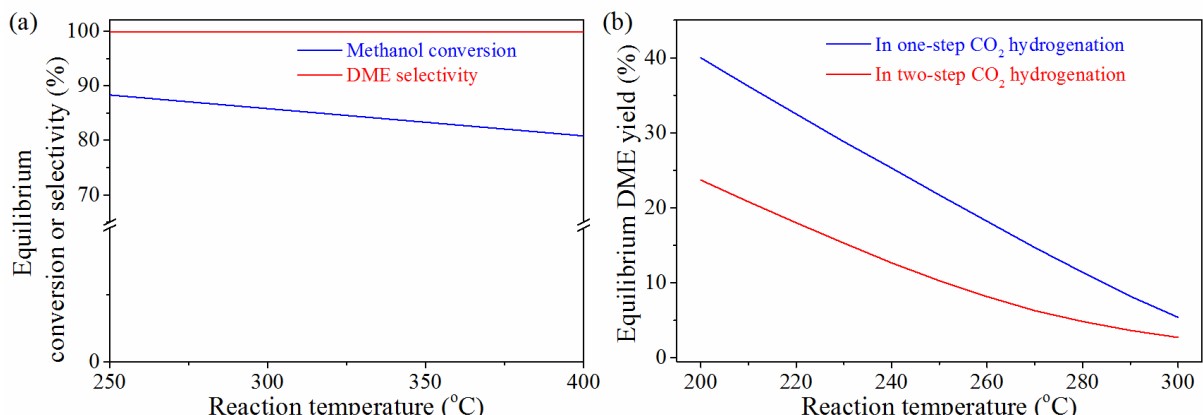

**Figure 3.** Thermodynamic results in the equilibrium reactions. (**a**) Equilibrium methanol conversion and DME selectivity in the methanol dehydration. (**b**) Equilibrium DME yield in the one-step or two-step $CO_2$ hydrogenation. Reproduced with permission from Shen et al. [35] and Palomo et al. [42], respectively published by Springer Nature, 2000, and Elsevier, 2019.

## 3. Copper-Based Catalysts

There is no doubt that Cu-based catalysts are the most studied in previous works. $Cu/ZnO/Al_2O_3$ (CZA) coupling with acidic sites, a well-known bifunctional catalyst, is firstly applied in CO hydrogenation to DME [43]. Subsequently, its activity in $CO_2$ hydrogenation to DME has been found and a great deal of study has been conducted. However, there is still no industrial application of Cu-based catalysts in $CO_2$ hydrogenation to DME, because the catalytic performance, e.g., DME selectivity, catalytic activity and stability, needs to be improved. To achieve this, investigating the nature of active sites and the reaction mechanism becomes necessary.

### 3.1. Active Sites

Previous work has already demonstrated that over the Cu-based catalyst, methanol is firstly formed on the Cu-based metallic sites, and DME is subsequently formed via dehydration of methanol on the acidic sites. In this case, the Cu sites play an important role in activating $CO_2$ and $H_2$. This is the initial step of $CO_2$ hydrogenation to DME and significantly affects the catalytic performance. In spite of its important role, the exact nature of Cu sites is still unclear. In an early study [44,45], the Cu-based catalyst becomes more active after the reduction in hydrogen flow. Thus, the metallic Cu ($Cu^0$) species are speculated to be the active sites for $CO_2$ hydrogenation. Further, temperature-programmed desorption of $N_2O$ ($N_2O$-TPD) is developed to exactly characterize the surface area of $Cu^0$ species, and the catalytic activity of $CO_2$ hydrogenation to methanol is found to linearly increase with increasing $Cu^0$ surface area [44,46–51]. Detailed results are shown in Figure 4a. Therefore, the researchers can declare that the $Cu^0$ species are responsible for the catalytic activity in $CO_2$ hydrogenation.

However, other researchers reported totally different results. Some publications [52–58] proposed that no reasonable relation could be shared between the $Cu^0$ surface area and the catalytic activity. Moreover, some authors even found that a too high $Cu^0$ surface area would inhibit the selectivity of methanol. The work of K. Stangeland et al. [59] indicated that the formation of methanol should not be attributed to the increasing $Cu^0$ surface area (Figure 4b), while the formation of CO was improved by the increasing $Cu^0$ surface area. Therefore, it is still doubtable that the $Cu^0$ species are the only active sites in $CO_2$ hydrogenation. These results made the researchers focus on other possibilities of the active nature over the Cu-based catalyst.

During $CO_2$ hydrogenation to DME, $Cu^0$ species are not the only detectable Cu-based species on the catalyst surface. Frei et al. [60] reported that $Cu^+$ species are also detected during the reduction of Cu-based catalysts, although most of them turned to $Cu^0$ species

after the reduction. This conclusion is agreed with in the work of Natesakhawat et al. [47] and Zhou et al. [48] However, other researchers propose [53,61–63] different results, e.g., the $Cu^+$ species can be found during and after the reaction of $CO_2$ hydrogenation. In this case, researchers started to pay attention to the role of $Cu^+$ species over the Cu-based catalysts in the $CO_2$ hydrogenation. Din et al. [53] confirmed the presence of $Cu^+$ species, and they also reported that the catalytic activity was dependent on the amount of neither $Cu^0$ nor $Cu^+$ species. Tan et al. [46] reported that although the catalytic activity increased with increasing $Cu^0$ surface area, no simple relation could be obtained when the $Cu^+$ species were involved. Furthermore, Sun et al. [61] detected both $Cu^0$ and $Cu^+$ species by Cu Auger spectra. They found that increasing catalytic activity was dependent on the amount of either $Cu^0$ or $Cu^+$ species; however, correlation of the catalytic activity with the amount of $Cu^+$ species was better than that with $Cu^0$ species. The results of Sun et al. [61], Dasireddy et al. [62] and Stangeland et al. [63] proposed that the enhanced catalytic activity was attributed to the total exposed Cu surface, and a higher molar ratio of $Cu^+/Cu^0$ species could also improve the catalytic activity (Figure 4c). Based on the notion that the $Cu^+$ species may also be active in $CO_2$ hydrogenation, Zhang et al. [64] and Pu et al. [65] employed $Cu_2O$ as the active phase and prepared $Cu_2O/ZnO$ and $Cu_2O/CeO_2$ catalysts, respectively. Their results demonstrate that the $Cu^+$ species can catalyze $CO_2$ hydrogenation to methanol; however, the catalytic activity of $Cu_2O$ supporting catalysts is lower than the typical CZA catalyst.

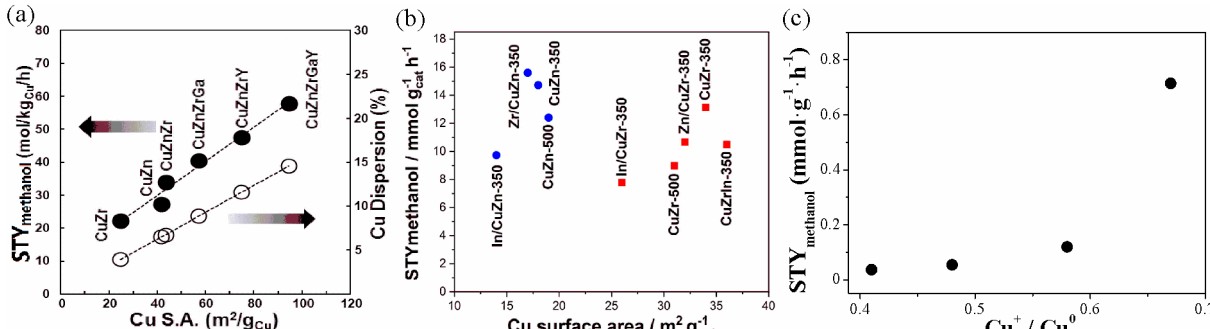

**Figure 4.** Correlation of space–time yield of methanol ($STY_{methanol}$) with (**a,b**) Cu surface area (S.A.) and (**c**) molar ratio of $Cu^+/Cu^0$. The Cu S.A. is measured by temperature-programmed desorption of $N_2O$. The ratio of $Cu^+/Cu^0$ is characterized by X-ray photoelectron spectroscopy. Reaction conditions: (**a**) $T$ = 240 °C, $P$ = 3 MPa, GHSV = 3000 mL·g$^{-1}$·h$^{-1}$, $H_2/CO_2$ = 3. (**b**) $T$ = 230 °C, $P$ = 3 MPa, GHSV = 3800 mL·g$^{-1}$·h$^{-1}$, $H_2/CO_2$ = 3. (**c**) $T$ = 200 °C, $P$ = 2 MPa, GHSV = 2000 mL·g$^{-1}$·h$^{-1}$, $H_2/CO_2$ = 3. Reproduced with permission from Natesakhawat et al. [47], Stangeland et al. [59] and Dasireddy et al. [62], respectively published by ACS Publications, 2012, Elsevier, 2021, and Elsevier, 2019.

As the catalytic activity of $CO_2$ hydrogenation is attributed to both the $Cu^0$ and $Cu^+$ species, researchers need to discuss the active nature of Cu-based catalysts considering both of them. Some researchers [55,66,67] confirmed the co-existence of $Cu^0$ and $Cu^+$ species on the catalyst surface, and the increasing amount of both $Cu^0$ and $Cu^+$ species favored the catalytic activity. Moreover, recent work [62,68] indicated that in $CO_2$ hydrogenation to methanol, the increasing $CO_2$ conversion was caused by a high number of $Cu^0$ species, while a high number of $Cu^+$ species improved the selectivity of methanol. Therefore, it is more acceptable that both $Cu^0$ and $Cu^+$ species are active in the $CO_2$ hydrogenation, and these species can be labeled as $Cu^{\delta+}$ with $0 \leq \delta < 2$.

Furthermore, researchers also focus on the effect of ZnO on the active nature and catalytic performance over the CZA catalyst, because ZnO is a necessary and important promoter in Cu-based catalysts. Results [69–73] of operando infrared spectroscopy (operando-IR) and density functional theory (DFT) calculations showed that $CO_2$ hydrogenation over the ZnO-supported $Cu^{\delta+}$ sites or the Cu/ZnO interface is more favorable than bulk $Cu^{\delta+}$ sites. This is because the activation energy for the formation of format intermediates is low-

ered due to the presence of ZnO. Moreover, ZnO is also found to improve the reducibility of Cu oxides [44,60], improve the dispersion of $Cu^{\delta+}$ species [44,69,73] and enhance the adsorption of $CO_2$ [70,72]. Some work [49,51] also proposed that the addition of $Al_2O_3$ enhanced the electron density of ZnO species, resulting in a stronger $CO_2$ adsorption ability. Although the Cu–ZnO interface and Cu–Zn alloy are also considered active sites in some publications [69,71], most of the results agree that the reaction mechanism is the same over $Cu^{\delta+}$ with or without ZnO. Therefore, the $Cu^{\delta+}$ species should be considered essential active sites for Cu-based catalysts in $CO_2$ hydrogenation.

In all, although the reduced Cu species have been confirmed as active sites, more efforts still need to be taken in future to further understand the exact function of $Cu^0$ and $Cu^+$ species. The effects of other important promoters that significantly improve the catalytic activity, e.g., $ZrO_2$ [44] and $Ga_2O_3$ [74], also need to be investigated. These efforts can raise our awareness on the active nature of Cu-based catalysts and are important for designing catalysts with high performance.

### 3.2. Reaction Mechanism

The reaction mechanism is another important topic for Cu-based catalysts. Different from the active nature, the researchers have agreed that there are two mechanisms in Cu-catalyzed $CO_2$ hydrogenation, the mechanism via format (HCOO*) intermediate and the other via CO* intermediate. Additionally, the mechanism via carbonyl (COOH*) intermediate is also proposed in a previous work [75]. However, the COOH* intermediate can easily turn into CO* and is further transformed to methanol. This indicates that in the mechanisms via COOH*, CO* is also a key intermediate. The mechanisms via COOH* and CO* are essentially the same.

The mechanism via HCOO* intermediate is considered the most important pathway in the Cu-catalyzed $CO_2$ hydrogenation to methanol [75–80]. This mechanism is confirmed by operando-IR [58,75,77,78,81], $^{14}CO_2$ [80]/$D_2$ [75,78] isotope experiments, kinetic study [79] and DFT calculations [58,76]. As shown in Figure 5, $CO_2$ is firstly hydrogenated to form HCOO*. Subsequently, one O atom in HCOO* is hydrogenolyzed and HCOO* is further hydrogenated to methoxyl ($CH_3O^*$) intermediate. Finally, the methanol is formed through the hydrogenation of $CH_3O^*$. Furthermore, HCOO* is found to be transformed to both methanol and CO.

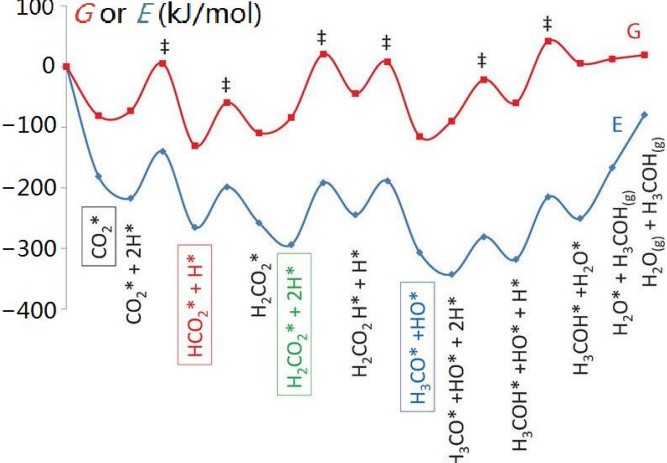

**Figure 5.** Energy (*E*) and standard Gibbs free-energy (*G*) pathway calculated for the $CO_2$ hydrogenation to methanol over $Cu/ZrO_2$. Reproduced with permission from Larmier et al. [81], published by John Wiley and Sons, 2017.

However, the mechanism via CO* intermediate cannot be totally ignored. Liu et al. [82] reported that both CO* and HCOO* were formed during $CO_2$ hydrogenation, but further conversion of HCOO* to methanol was more difficult than CO*. Therefore, they believed

that the reaction mechanism of $CO_2$–CO–methanol was more reasonable. Moreover, some results of DFT calculations [71,83,84] and kinetic study [83] also confirm the mechanism via CO*. Thus, researchers also studied the co-existence of the two mechanisms. Yu et al. [85] found that the $Cu^0$ species catalyzed the formation of methanol via HCOO*, while the RWGS reaction also occurred. Subsequently, the formed CO could be transferred to the $Cu^+$ species and be further hydrogenated to methanol. Their results indicated that both the $CO_2$-to-methanol and RWGS reactions were catalyzed by the $Cu^0$ species, and this is also confirmed by Kunkes et al. [86] Moreover, Poto et al. [87] reported that the reaction conditions could affect the reaction mechanism. Particularly, the mechanism via CO* was more favorable with high reaction temperature and high $H_2/CO_2$ ratio. With decreasing temperature and stoichiometric ratio of $H_2/CO_2$, the mechanism via HCOO* became more important. These results can explain why in a previous study under the typical conditions (<240 °C, $H_2/CO_2$ = 3) for $CO_2$ hydrogenation to methanol, the mechanism via HCOO* was speculated to be principal.

### 3.3. Stability

Cu-based catalysts often suffer from poor stability. Three reasons cause this problem [28]: (i) the metallic Cu can be easily sintered at quite a low temperature of 240 °C; (ii) the coke formation can deactivate the metallic and acidic sites; (iii) the formed $H_2O$ in the reaction may oxidize the $Cu^0$ species and deactivate the acidic sites of zeolites. To overcome this problem, $ZrO_2$ is added as a promoter for the CZA catalysts [88], in order to slow the oxidation of $Cu^{\delta+}$ species. Modification of acidic sites is another important way to improve the stability [89]. Moreover, techniques of regeneration are also employed in the $CO_2$ hydrogenation procedure to prolong the lifetime of catalysts. The corresponding results are summarized by Sobczak et al. [28].

However, in $CO_2$ hydrogenation, more water is formed than in CO hydrogenation due to the RWGS reaction. Especially at increasing reaction temperature, the RWGS reaction is improved and even more water is formed, resulting in a serious deactivation of Cu-based catalysts. High reaction temperature also leads to sintering of the reduced Cu species. Therefore, to improve the catalytic activity together with the stability in one-step $CO_2$ hydrogenation to DME, it is necessary to design a more stable active phase for $CO_2$ hydrogenation to DME at high reaction temperatures.

## 4. Precious Metal Catalysts

Precious metals have been widely employed in heterogeneous catalysis due to their enriched unoccupied d-bands favoring the activation of small molecules such as CO, $CO_2$, $H_2$, etc. In the $CO_2$ hydrogenation reaction, palladium is found to be active for hydrogenating $CO_2$ to methanol. Thus, one-step $CO_2$ hydrogenation to DME can be realized by coupling a Pd-based active phase and the acidic phase.

Most of the related publications [90–93] agree that the reduced Pd species ($Pd^0$) are active sites for $CO_2$ hydrogenation. However, agreement cannot be reached in the case of reaction mechanism. Bonivardi et al. [94] reported that HCOO* was the key intermediate over the $Pd/Ga_2O_3$ catalyst, and they also proposed a possible reaction pathway of $HCO_3$ * (or $CO_3$ *) $\rightarrow$ HCOO* $\rightarrow$ HCOOH* $\rightarrow$ $CH_3O$* $\rightarrow$ $CH_3OH$*. Moreover, the works of Schild et al. [95] and Arunajatesan et al. [96] also confirm the HCOO* mechanism. On the contrary, Malik et al. [97] proposed that over CaO–Pd/ZnO, $CO_2$ was hydrogenated via the CO* intermediate, and similar results were obtained over the $Pd/SiO_2$ catalyst in the work of Cabilla et al. [98].

Based on the current results, we can speculate that this difference is caused by the nature of active sites. As presented in Figure 6, on one hand, over the Pd-based catalysts with the oxides favorably activating the $CO_2$ molecular, e.g., $Ga_2O_3$ [94], $Al_2O_3$ [96] or $ZrO_2$ [95], the $CO_2$ molecule is activated by the oxides but not by metallic Pd, according to the operando-IR results. In this case, HCOO* is characterized as the key intermediate. A typical case is proposed by Cardona-Martinez et al. [91], e.g., over the $Pd/Ga_2O_3$ cata-

lyst, the metallic Pd sites are responsible for activating $H_2$ and $Ga_2O_3$ is responsible for activating $CO_2$. Subsequently, the activated $CO_2$ can be transferred to the Pd surface and be hydrogenated to methanol. On the other hand, when more inert oxides, e.g., $SiO_2$ [98] or ZnO [97], are employed into Pd-based catalysts, $CO_2$ is adsorbed and activated by $Pd^0$ species and the RWGS mechanism is dominated, indicating that on the active sites of $Pd^0$ species, $CO_2$ hydrogenation occurs via the CO* intermediate. Therefore, if only Pd is involved as the active site, $CO_2$ is firstly hydrogenated to CO then further converted to methanol. This is different from Cu- and oxide-based catalysts with formate mechanisms.

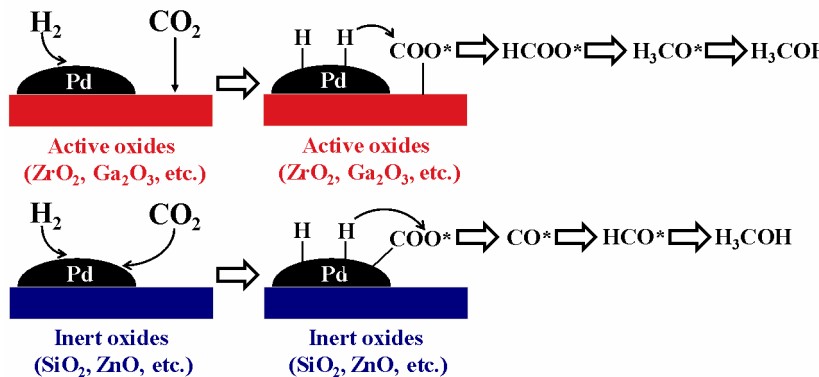

**Figure 6.** Schematic reaction mechanism of $CO_2$ hydrogenation over Pd-based catalysts with addition of different oxides.

In Table 2, typical examples of Cu- and Pd-based catalysts with high performances in $CO_2$ hydrogenation to DME are presented. In the case of Cu-based catalysts, it is observable that different promoters significantly affect the catalytic activity and DME selectivity. The addition of $La_2O_3$ into the CZA/HZSM-5 catalyst clearly enhanced the formation of DME [99]. The utilization of $ZrO_2$ [100,101] or $Ga_2O_3$ [74] to replace $Al_2O_3$ in the CAZ catalyst improves the space–time yield of DME ($STY_{DME}$); however, the effect of $CeO_2$ is less significant [102]. Moreover, the catalysts with ferrierite as their acidic sites exhibit better catalytic performance than those with HZSM-5 [100,103]. In the case of the Pt-based catalysts, the previous study indicates that addition of basic promoters such as CaO [90,98] and oxide supports such as $Ga_2O_3$ [91,94] or $CeO_2$ [97] can improve the catalytic activity, because the adsorption and activation of $CO_2$ can be enhanced. Some other works [104,105] also reported catalysts with a bimetallic active phase of Cu–Pd. However, the mixture of two kinds of active metal causes difficulties in understanding the active nature of the catalysts. Moreover, compared with the mono-metal-based catalysts (Table 2), the Cu–Pd bimetallic catalysts exhibit no significant enhancement in the catalytic performance.

It is also observable in Table 2 that the catalytic performance is clearly affected by the reaction conditions. Irrespective of the composition of catalysts, $STY_{DME}$ is improved by the increasing reaction pressure. Higher GHSV could enhance the DME selectivity and inhibit the $CO_2$ conversion for Cu-based [100,101] and Pd-based catalysts [90]. However, the reaction temperature exhibits different effects on different catalysts. Over the CZA/HZSM-5 catalysts, the increase in the reaction temperature leads to a decreasing DME selectivity and a decreasing $STY_{DME}$ [106]. However, different from Cu-based catalysts, at a higher reaction temperature of 325 °C, the Ca-Pd/$CeO_2$ catalyst exhibits a higher $STY_{DME}$ [90]. Moreover, R. Chu et al. [93] reported that the Pd/HZSM-5 catalyst exhibited a much better sulfur tolerance than the CZA/HSAM-5 catalyst (Figure 7).

**Table 2.** Catalytic performance of Cu- and Pd-based catalysts in the $CO_2$ hydrogenation to dimethyl ether (DME).

| Catalyst | $T$ (°C) | $P$ (MPa) | GHSV (mL·g$^{-1}$·h$^{-1}$) | $CO_2$ Conversion (%) | DME Selectivity [1] (%) | STY$_{DME}$ [2] (mmol·g$^{-1}$·h$^{-1}$) | Reference |
|---|---|---|---|---|---|---|---|
| Cu–ZnO–Al$_2$O$_3$/HZSM-5 | 225 | 3.0 | 48,000 | 2.9 | 75.6 | 5.0 | [106] |
| Cu–ZnO–Al$_2$O$_3$/HZSM-5 | 260 | 3.0 | 1500 | 19.2 | 91.6 | 1.7 | [107] |
| Cu–ZnO–Al$_2$O$_3$/HZSM-5 | 300 | 3.0 | 1800 | 29.5 | 24.1 | 0.92 | [108] |
| Cu–ZnO–Al$_2$O$_3$/Ferrierite | 250 | 3.0 | 12,000 | 19.8 | 79.0 | 6.2 | [103] |
| Cu–ZnO–CeO$_2$/Ferrierite | 260 | 3.0 | 8800 | 14.0 | 71.6 | 5.0 | [102] |
| Cu–ZnO–ZrO$_2$/Ferrierite | 260 | 5.0 | 8800 | 26.0 | 81.3 | 7.2 | [100] |
| Cu–ZnO–ZrO$_2$/HZSM-5 | 240 | 3.0 | 10,000 | 13.4 | 80.8 | 4.9 | [101] |
| La–Cu–ZnO–Al$_2$O$_3$/HZSM-5 | 250 | 3.0 | 3000 | 43.8 | 94.3 | 9.1 | [99] |
| Cu–ZnO–Ga$_2$O$_3$/HZSM-5 | 260 | 3.0 | 1500 | 22.3 | 89.4 | 2.8 | [74] |
| CaO–Pd/CeO$_2$ | 325 | 2.0 | 2400 | 30.9 | 70.5 | 5.3 | [90] |
| Pd/Ga$_2$O$_3$/Nb$_2$O$_5$ | 270 | 1.7 | 2700 | 5.0 | 53.0 | 0.39 | [91] |
| CeO$_2$–CaO–Pd/HZSM-5 | 250 | 3.0 | 1600 | 21.7 | 67.3 | 1.6 | [93] |
| Cu/ZnO–Pd/CNT/HZSM-5 | 250 | 5.0 | 25,000 | 18.9 | 50.3 | 0.58 | [104] |
| Pd–Cu–Al$_2$O$_3$/HZSM-5 | 260 | 2.0 | 3000 | 11.3 | 53.6 | 0.33 | [105] |

[1] Selectivity of DME in the products without CO. [2] STY$_{DME}$ is the space–time yield of DME.

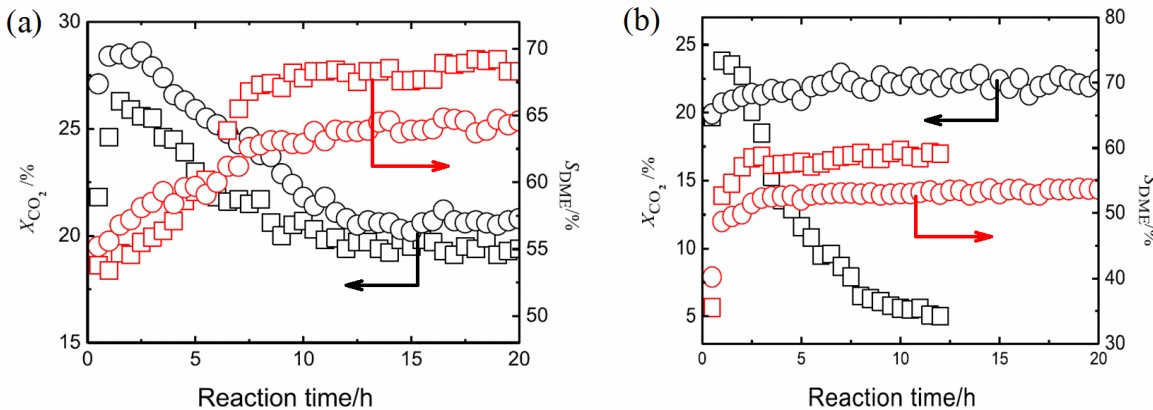

**Figure 7.** $CO_2$ conversion ($X_{CO2}$) and selectivity of dimethyl ether ($S_{DME}$) with (square) and without (circle) $H_2S$ in the gas feed. (**a**) CaO–CeO$_2$–Pd/HZSM-5; (**b**) Cu–ZnO–Al$_2$O$_3$/HZSM-5. Conditions: 250 °C, 3 MPa, 1600 mL·g$^{-1}$·h$^{-1}$, $H_2$/$CO_2$ = 4.4. Reproduced with permission from Chu et al. [93], published by Elsevier, 2017.

## 5. Indium- and Gallium-Based Catalysts

### 5.1. Indium-Based Catalysts

$In_2O_3$ is found to be active in $CO_2$ hydrogenation to methanol and a methanol selectivity of 80% is detectable even at a reaction temperature of >300 °C [109,110]. Researchers propose that the oxygen vacancies on defective $In_2O_3$ are active sites for $CO_2$ hydrogenation. The DFT calculations [110,111] demonstrate that $CO_2$ is activated over $In_2O_3$ via oxidation of the oxygen vacancies. After $CO_2$ hydrogenation, the oxygen vacancies are reproduced by the reduction of $H_2$ or CO. Moreover, results also confirm that the HCOO* species are the intermediate for $CO_2$ hydrogenation over $In_2O_3$ catalysts.

Based on these facts, Pechenkin et al. [31] prepared a bifunctional catalyst of $In_2O_3$/halloysite nanotubes to catalyze $CO_2$ hydrogenation. Their results (Figure 8) indicate that the DME selectivity reaches 65% at 200 °C; however, the increasing reaction temperature inhibits DME selectivity and improves methanol selectivity. At 300 °C, DME selectivity is only 20%. The authors declared that the exothermicity of the methanol dehydration inhibited the formation of DME with increasing reaction temperature. Therefore, in spite of the high stability, the catalytic activity and DME selectivity over the oxide-based catalysts are still low with increasing reaction temperature.

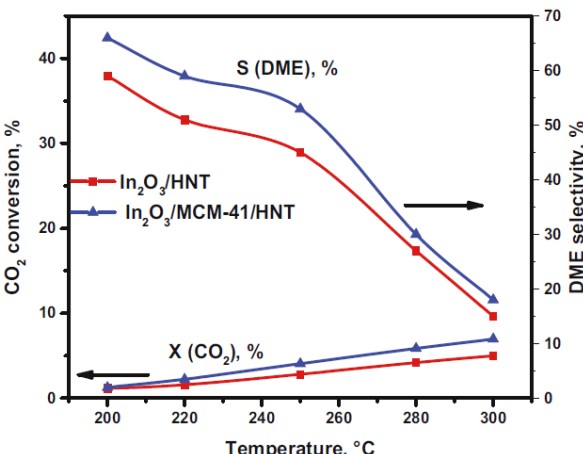

**Figure 8.** Catalytic performance over the $In_2O_3$/halloysite nanotubes catalysts at *T* = 200–300 °C, *P* = 4 MPa and GHSV = 12,000 mL·g$^{-1}$·h$^{-1}$. Reproduced with permission from Pechenkin et al. [31], published by De Gruyter, 2021.

*5.2. Gallium-Based Catalysts*

As presented before in the work of Collins et al. [94], $Ga_2O_3$ is employed as the active site for activating $CO_2$ over Pd/$Ga_2O_3$. Moreover, their results of operando-IR indicate that the methanol is formed via the format intermediate, the same as the Cu-based catalysts. It is worth noting that the presence of Pd in the catalyst does not change the reaction mechanism; it only improves the activation of $H_2$ and the catalytic activity. Therefore, $Ga_2O_3$ can independently play a role in the activation and hydrogenation of $CO_2$. Moreover, based on the results of the theoretical simulation, Studt et al. [112] prepare a Ni/$Ga_2O_3$/$SiO_2$ catalyst for $CO_2$ hydrogenation to methanol. Results indicate that the Ni/$Ga_2O_3$/$SiO_2$ catalyst with Ni/Ga = 5/3 exhibits a similar yield of methanol as the CZA catalyst. This is because the theoretical simulation indicates that oxygen adsorption energy is closed over Ni/$Ga_2O_3$/$SiO_2$ and CZA catalysts. The authors propose that the oxygen adsorption energy is related to the adsorption and activation of $CO_2$ and further affects the selectivity. The addition of $Ga_2O_3$ enhances the oxygen adsorption energy as well as the methanol selectivity, due to the increasing ability of $CO_2$ adsorption.

Therefore, the two above examples indicate the unique propriety of $Ga_2O_3$ for the adsorption and activation of $CO_2$. Further, another compound of gallium, GaN, is also employed as an active phase for $CO_2$ hydrogenation in the work of Liu et al. [33] The GaN catalyst exhibits a high DME selectivity under the conditions of *T* = 360 °C, *P* = 2.0 MPa, GHSV = 3000 mL·g$^{-1}$·h$^{-1}$, and $H_2$/$CO_2$ = 2. Furthermore, with decreasing contact time, the DME selectivity increases while the methanol selectivity decreases, indicating that in $CO_2$ hydrogenation over GaN, DME is the primary product and methanol is the secondary product. The results of surface reaction further confirm DME as the primary product, while CO is also speculated as another primary product. As shown in Figure 9, the DFT calculations indicate that $CO_2$ can be activated by different modes on the different GaN crystalline planes of (100) and (110). The following hydrogenation of $CO_2$ leads to the formation of $CH_3$ * and HCOO* intermediates, and the final formation of DME occurs on

the interface of (100)/(110). Therefore, the reaction pathway over GaN is quite different from the metallic catalyst, indicating that GaN could be a totally new kind of active phase for $CO_2$ hydrogenation to DME.

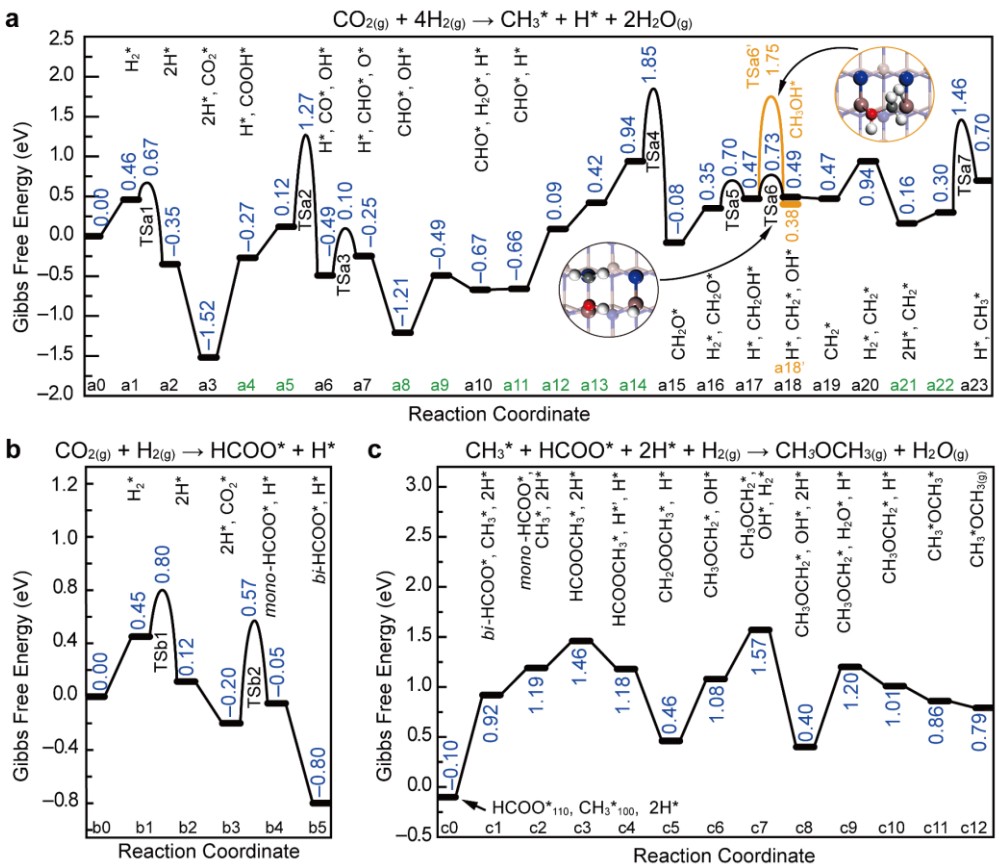

**Figure 9.** The density functional theory calculation results of $CO_2$ hydrogenation over GaN. (**a**) Gibbs free energy diagram of the $CO_2$ hydrogenation to methyl ($CH_3^*$) on the GaN(100) plane. (**b**) Gibbs free energy diagram of the $CO_2$ hydrogenation to formate ($HCOO^*$) on the GaN(110) plane. (**c**) Gibbs free energy diagram for the coupling of $HCOO^*$ and $CH_3^*$ to $CH_3OCH_3^*$ on the (110)/(100) interface. TS: transition states. Reproduced with permission from Liu et al. [33], published by Springer Nature, 2021.

## 6. Transition Metal Oxide Catalysts

Different from the metallic species, metal oxides should not be sintered even at high temperatures of around 400 °C. Thus, enhanced stability can be expected over the metal oxide-based catalysts. Liu et al. [32] proposed a $ZnAlO_x$ catalyst prepared by the co-precipitation method (Figure 10). Under the reaction conditions of $T = 320$ °C, $P = 3$ MPa and GHSV = 12,000 mL·g$^{-1}$·h$^{-1}$, the DME selectivity in the products without CO can reach ca. 50% with a $CO_2$ conversion of 5.2%. The active phase is speculated to be the oxygen vacancies over the crystallite of $ZnAl_2O_4$. Moreover, the results of in-situ IR spectroscopy demonstrate that $CO_2$ is hydrogenated to methanol via the $HCOO^*$ and $CH_3O^*$ species over the oxygen vacancies. Other authors [113] reported that the –OH groups on the surface of $ZnAlO_x$ also play an important role in $CO_2$ hydrogenation.

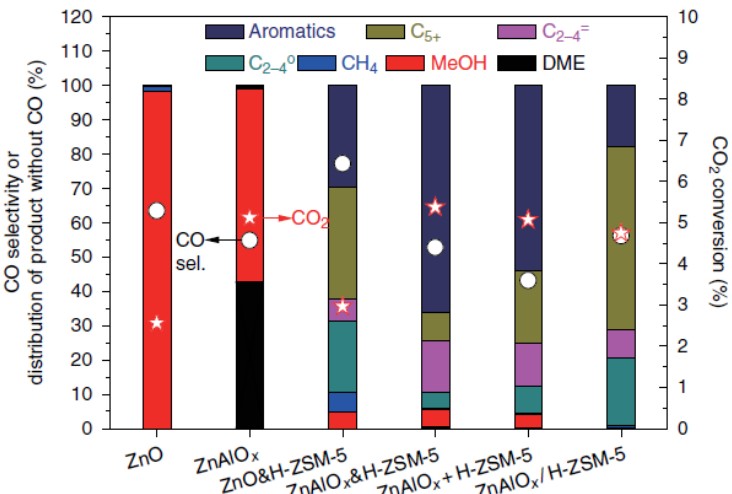

**Figure 10.** Catalytic performance over the $ZnAlO_x$ catalyst at $T$ = 320 °C, $P$ = 3 MPa and GHSV = 12,000 mL·g$^{-1}$·h$^{-1}$. Reproduced with permission from Ni et al. [32], published by Springer Nature, 2019.

However, prepared by the same method of co-precipitation and evaluated under a higher reaction temperature of 370 °C and a slightly lower GHSV of 10,800 mL·g$^{-1}$·h$^{-1}$, the $ZnAl_2O_4$ catalyst exhibits a DME selectivity of only 15% in the work of Wang et al. [113] Moreover, Song et al. [114] reported that the $ZnAl_2O_4$ catalyst, also prepared by the co-precipitation method, exhibited a methanol selectivity of >95% and a DME selectivity of <2%, under the conditions of $T$ = 280 °C, $P$ = 3 MPa and GHSV = 1800 mL·g$^{-1}$·h$^{-1}$. Therefore, it is uncertain whether $ZnAl_2O_4$ is an effective catalyst for the $CO_2$ hydrogenation to DME, although it exhibits good stability in all the mentioned publications. It is necessary to find out the reason why the selectivity is so different regarding fact that the preparation method is the same and the reaction conditions are similar.

## 7. Acidic Sites and the Synergetic Effect

In the bifunctional catalysts, acidic sites play a role in converting methanol to DME (Figure 11). Thus, the properties of the acidic sites and the coupling of metallic and acidic sites significantly affect the catalytic activity, the DME selectivity and the stability. In a CAZ/acids catalyst [115,116], $\gamma$-$Al_2O_3$ is employed as an acidic site for CO hydrogenation to DME and a high selectivity can be obtained. However, for $CO_2$ hydrogenation to DME, $\gamma$-$Al_2O_3$ is no longer a feasible acidic site. This is because the increasing accumulation of water in the $CO_2$ hydrogenation, due to the RWGS reaction, leads to a serious deactivation of $\gamma$-$Al_2O_3$ with low resistance to water [27]. Therefore, the hydrophobic zeolites become the most used acidic sites in one-step $CO_2$ hydrogenation to DME due to their good resistance to water.

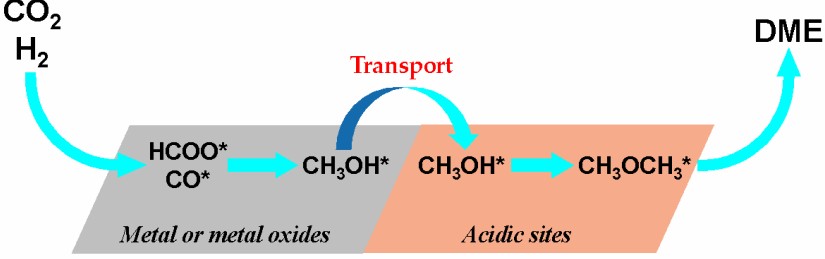

**Figure 11.** Scheme for the reaction pathway of the one-step $CO_2$ hydrogenation to DME.

Different kinds of zeolites have been employed as acidic sites, e.g., ZSM-5 [117,118], SAPO-11 [119], mordenite [120], ferrierite [100], etc. Recently, researchers focused on

the effect of composition, textural properties and acidity on the catalytic performance. Frusteri et al. [121] reported that the ratio of Si/Al affected the selectivity of products and the stability through changes in the amount of acid and the resistance to water. A high Si/Al ratio enhances the resistance to water and the stability, but too high a Si/Al significantly lowers the surface acidity and catalytic activity. A low Si/Al ratio leads to high activity due to the increasing acidity, but the catalytic stability becomes poor. Further, by adjusting the Si/Al ratio from 27 to 127, the authors also found that over the CZA/HZSM-5 catalyst, the zeolite with Si/Al = 38 was optimized for stable hydrogenation of $CO_2$ to DME with high DME selectivity.

There is no doubt that the acidity of zeolites plays a key role in the catalytic activity and DME selectivity. There are several parameters to characterize the acidity of zeolites [122–126], e.g., the amount of weak/strong acidic sites, the strength of acidic sites and the type (Bronsted or Lewis) of acidic sites. Typically, temperature-programmed desorption of $NH_3$ is employed to characterize the amount and strength of acidic sites, and the in-situ IR spectroscopy of pyridine adsorption is employed to characterize Bronsted and Lewis acidic sites. In the case of the acidic type, both Bronsted and Lewis acidic sites are active sites for the dehydration of methanol to DME [27]; however, the Bronsted acidic sites also favor the dehydration of methanol to HCs [127]. Moreover, the Bronsted acidic sites also cause the formation of carbon residuals, resulting in the deactivation of catalysts. Regarding the strength, the increasing strength of acids can improve the catalytic acidity; however, acidic sites that are too strong catalyze the secondary conversion of DME and inhibit DME selectivity. Regarding the amount, an increase in the amount of weak Lewis acidic sites is found to be favorable in enhancing the catalytic performance in $CO_2$ hydrogenation to DME.

Moreover, the porosity of zeolites also affects the catalytic performance [128,129]. The increasing distribution of mesopores with diameters of 10–50 nm is reported to favor the stability by inhibiting the formation of carbon residuals. However, the increasing distribution of micropores with diameters of smaller than 2 nm leads to increased formation of carbon residuals, resulting in deactivating the zeolites, although more micropores can improve the surface area of zeolites and the catalytic activity.

## 8. Conclusions and Research Directions

To conclude, one-step $CO_2$ hydrogenation to DME is a promising process for the efficient utilization of $CO_2$. However, the reasonable design of efficient catalysts is still a challenge. This paper summarized the recent process in heterogeneous catalysis for $CO_2$ hydrogenation to DME, and focused on the nature of active sites and the reaction mechanism over different kinds of catalysts. In the case of Cu-based catalysts, it is generally agreed that the reduced metallic Cu species are the active sites, although the function of $Cu^+$, $Cu^0$ and $Cu^{\delta+}$ species is still unsure. The formate mechanism is dominated over the Cu-based catalysts, while the RWGS mechanism can be improved with increasing reaction temperature and increasing partial pressure of $H_2$. The presence of ZnO promoter improves the adsorption and activation of $CO_2$, thus favoring the catalytic activity. However, ZnO should not affect the reaction mechanism in $CO_2$ hydrogenation to DME. Precious metal-based catalysts can also catalyze $CO_2$ hydrogenation to DME. Metallic Pd and the Pd–Cu alloy are the most used active phases. Different from the Cu-based catalysts, The RWGS mechanism is dominated over the metallic Pd sites. However, with reference to the high cost, the catalytic performance over Pd-based catalysts is not significantly enhanced. The $In_2O_3$- and $ZnAl_2O_4$-based catalysts exhibit an excellent stability at high reaction temperatures. The oxygen vacancies are reported as the active sites and the formate species are speculated to be key intermediates. However, the catalytic activity and DME selectivity need to be improved. Over the GaN-based catalyst, DME is formed as a primary product while methanol is a secondary product. This is totally different from the metallic or oxide-based catalysts. The study of reaction mechanism indicates that both HCOO* and COOH* are key

intermediates during the formation of DME. Based on current understanding, perspectives on the important research directions are proposed as follows:

(i)    Because of the RWGS reaction, CO is always formed as a by-product during $CO_2$ hydrogenation, which lowers the selectivity of the high-value-added target products. Therefore, a deep understanding on the activation modes and subsequent transformation paths of $CO_2$ molecules is very important for the rational design of a catalyst with limited CO selectivity. In the case of selective $CO_2$ hydrogenation to DME, integrating the rich and available knowledge of the elementary steps into finding/re-optimizing the component and structure of catalysts via emerging techniques such as artificial intelligence is promising for inhibiting side reactions, including RWGS, which can be an important direction for the advancement of high-performance catalysts;

(ii)    In view of the exothermal nature of $CO_2$ hydrogenation to DME and the easy sintering of metallic Cu, more stable Cu-based catalysts for the selective synthesis of DME are expectable at lower reaction temperatures, which brings on a great challenge for improving the catalytic activity. Indeed, the active site of Cu-based catalysts for both CO and $CO_2$ hydrogenation reactions is still not unambiguously resolved. In this case, taking the redox nature of $CO_2$ hydrogenation reactions into account, the disclosure of the dynamically evolved oxidation number of Cu and possible redox cycles between $Cu^0$ and $Cu^+/Cu^{\delta+}$ under the reaction conditions is crucial in identifying the active site. Thus, keeping these notions in mind, molecular-level understandings of $CO_2$ hydrogenation to DME are reasonably expected with the development/modification of in-situ/operando techniques, which can be the second direction of the themed topic;

(iii)    In spite of their enhanced stability, catalytic activity and DME selectivity need to be improved for oxide-based catalysts. Previous works have consistently revealed the important role of oxygen vacancies. However, no simple correlation can be made between the amount of oxygen vacancies and the catalytic activity/DME selectivity. Thus, mechanistic studies on the formation/consumption of oxygen vacancies and the kinetics of the oxygen vacancies participating in elementary steps during $CO_2$ hydrogenation are necessary works for the rational design and advancement of oxide-based catalysts for $CO_2$ hydrogenation to DME with improved catalytic performance;

(iv)    Our results indicate that GaN itself is an effective bifunctional catalyst for the selective synthesis of DME from $CO_2$ hydrogenation. More importantly, the unique mechanism, i.e., the direct hydrogenation of $CO_2$ to DME without the formation of the methanol intermediate, is consolidated by the results of the catalytic experiments, operando spectroscopy, and DFT calculations. Indeed, the large-scale application of GaN as the catalyst is inevitably limited by the expensive Ga source. However, with the understanding of the wurtzite structure and the formation of DME over GaN cooperated by the two facets of (100), (110) and their interfaces, the mechanistically guided exploration of this novel, cheap bifunctional catalyst is attractive, which is worthy of being another research direction;

(v)    The summarized catalytic results indicate that one-step synthesis of DME via $CO_2$ hydrogenation is a very promising candidate for CCSU. Moreover, the physical and chemical properties of DME in comparison with those of methanol make an attractive DME economy either parallel with or supplementary to the methanol economy. In both cases, however, investigations on both the expanded application of DME and the extension of DME-derived high-value-added chemicals/materials are required in addition to the life cycle analyses of the whole process. These interdisciplinary works and process systems engineering are important for the establishment of the DME economy, which is worthy of being considered a long-term research direction.

**Author Contributions:** Writing and editing—original draft preparation, C.L.; writing—review and supervision, Z.L. All authors have read and agreed to the published version of the manuscript.

**Funding:** This work is financed by the National Natural Science Foundation of China (21603135), the Fundamental Research Funds for the Central Universities (GK201901001) and the Natural science basic research program of Shaanxi province (2021JQ-306).

**Data Availability Statement:** Not applicable.

**Conflicts of Interest:** The authors declare no conflict of interest.

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
