# Peer review of "Perspective on CO2 Hydrogenation for Dimethyl Ether Economy"

_catalysts, doi:10.3390/catal12111375_

Round 1

Reviewer 1 Report

This review was well-written by the professionals in carbon dioxide conversion. It presents its own focus on this area of research, which distinguishes it from other review papers. The review is not overloaded with research details, so it is easy to understand. The information will be understandable and accessible to non-specialists in this field. 

It deserves respect that there are references to other reviews. 

The authors tried to describe the achievements of recent years, in particular, they draw the attention of researchers to a still little studied but promising GaN catalyst.

I was surprised by almost no typos.

Page 8. Lines 303-304. "CO hydrogenation" should be replaced by "CO2 hydrogenation".

Page 13. Line 462. "in situ" should be written by the slanted font.

Reviewer 2 Report

The manuscript presented is a review of the hydrogenation of carbon dioxide to dimethyl ether. The subject is important and of great scientific interest. The manuscript is very well organized and easy to follow. However, there are typographical errors that should be corrected. I recommend this manuscript to be published in Catalysts, some issues should be addressed.

·         Considering the three main groups of catalysts: copper-based catalysts; precious metal catalysts; metal oxide and other catalysts. The last one was less developed and same improvements are needed.

·         The work developed with indium-based and gallium-based catalysts have been grown in the last years, the suggestion is to create an independent section for these two metals. Is it possible to improve this section with some mechanistic considerations in this type of metal-based catalysts?

·         The section for other metal oxide catalysts should be other one.

·         The section acidic sites and the synergetic effect are almost based in zeolite catalysts, the title of this section should be reconsidered. A figure to illustrate the acidic sites importance in this reaction would improve the manuscript. 

·         When the authors cite another author in the manuscript the last name is enough.  

Reviewer 3 Report

Comments and suggestions are given in the attached document. 
